# Diagnostic value of ASVS for insulinoma localization: A systematic review and meta-analysis

**Hao Wang**⬤*°, **Ying Ba**°, **Qian Xing, Run-Ce Cai**

Department of Endocrinology, The First Affiliated Hospital of Dalian Medical University, Dalian, Liaoning Province, China

° These authors contributed equally to this work.

* wanghaodl@126.com

**Data Availability Statement:** All relevant data are within the manuscript and its Supporting Information files.

**Funding:** The authors received no specific funding for this work.

## Abstract

### Background

Previous studies on the diagnostic value of arterial calcium stimulation with hepatic venous sampling (ASVS) for the localization of insulinoma have reported inconsistent results. Here, we performed a meta-analysis of the relevant published studies.

### Methods

PubMed, Embase, Web of Science, the Cochrane Library, and Wanfang Data were searched for studies on the diagnostic value of ASVS in insulinoma localization published up to May 2019. We calculated the sensitivity, specificity, positive likelihood ratio (PLR), negative likelihood ratio (NLR), diagnostic odds ratio (DOR), and receiver operating characteristic (ROC) curve of ASVS in the localization of insulinoma.

### Results

We included ten studies involving 337 patients in the study. The pooled sensitivity, specificity, PLR, and NLR were 0.93 (95% confidence interval [CI]: 0.83–0.97), 0.86 (95%CI: 0.75–0.93), 6.8(95%CI: 3.7–12.7), and 0.08 (95%CI: 0.03–0.19), respectively. The DOR was 84 (95%CI: 30–233), and the area under the ROC curve was 0.96 (95%CI: 0.94–0.97).The results of the heterogeneity of the studies (P = 0.00, $I^2$ = 80.17) were calculated using forest plots of the DOR.

### Conclusion

ASVS is of significant value in localization of insulinoma. If a qualitative diagnosis of insulinoma is definite and the imaging examination results are negative, ASVS should be performed to confirm the localization of insulinoma.

**Competing interests:** The authors have declared that no competing interests exist.

## Introduction

Although insulinoma is the most common pancreatic cell tumor, it remains a rare disease with an annual incidence of 1/1,000,000 to 10/1,000,000 [1]. Insulinoma is characterized by the Whipple triad, with varying degrees of hypoglycemia, and severe conditions may be life-threatening. Insulinoma is usually treated with surgery [2,3]. Insulinomas are usually small in size, mostly <2 cm in diameter [4]. Intraoperative ultrasound has a very high positive detection rate, so some researchers believe that preoperative localization is unnecessary due to the possibility of intraoperative ultrasound localization of this tumor [5]. However, most researchers disagree with that opinion and believe that a definite preoperative location will help reduce surgical trauma, reduce the operation time, ensure a successful operation, and reduce surgeon and patient worries. In addition, definite preoperative localization and diagnosis can facilitate complete tumor resection and reduce the incidence of complications such as pancreatic fistula. Therefore, preoperative localization is highly recommended[6–9].

Methods for localizing insulinoma are generally divided into non-invasive and invasive tests. Computed tomography (CT), magnetic resonance imaging (MRI), and other non-invasive tests do not have high positive rates, i.e.31–57% [10,11], and the positive rate of digital subtraction angiography is 65–83% [11,12]. Endoscopic ultrasonography (EUS) is a mildly invasive test with a positive rate of 80–89% [13,14], depending greatly on the operator's technique and experience. The above tests can detect insulinoma only morphologically, but arterial calcium stimulation with hepatic venous sampling (ASVS) can locate insulinoma functionally. ASVS is performed by selectively catheterizing the gastroduodenal artery, the superior mesenteric artery, and the proximal and distal splenic arteries; rapidly injecting calcium gluconate into these sites for stimulation; and collecting blood samples from the hepatic veins at different time points before and after the challenge. The ratios of the stimulation values to basic values are calculated, with the highest ratio treated as the peak ratio. The pancreatic region supplied by the artery with the highest peak ratio is considered the region where the tumor is located. The accuracy of localization does not depend on the tumor size, and even occult insulinoma can be detected. Therefore, localization of insulinoma by ASVS should be very accurate. However, there has been no meta-analysis of ASVS to date; accordingly, we performed the present meta-analysis to evaluate the diagnostic value of EUS for localization of insulinoma.

## Methods

### Data acquisition, search strategy, and inclusion and exclusion criteria

We followed the Preferred Reporting Items for Systematic Reviews and Meta-Analyses (PRISMA) guidelines[15]. PubMed, Embase, Web of Science, the Cochrane Library, and Wanfang Data were searched for studies on the value of ASVS in insulinoma localization and diagnosis published before May 2019. The following search terms were used: arterial calcium stimulation, insulinoma, localization, and pancreatic tumor. The search strategies were as follows: PubMed: ("arterial calcium stimulation" AND "insulinoma" [MeSH]); Embase: Emtree term–exploded = (insulinoma AND arterial calcium stimulation); Web of Science: TS = (insulinoma AND arterial calcium stimulation); the Cochrane Library and Wanfang: keyword = (insulinoma AND arterial calcium stimulation). Additional publications in the reference lists and citation sections of recovered articles were also searched. There were no restrictions on publication status or publication language.

Two authors of this study, Hao Wang and Ying Ba, independently searched the literature and included relevant research. To settle any disagreement, a third author (Qian Xing) joined the discussion to resolve the issue. The inclusion criteria for the study were as follows:(1)

Studies reported on patients with clear diagnosis of hypoglycemia and suspected insulinoma. (2) The diagnosis of insulinoma was confirmed by postoperative pathology or by the combination of clinical manifestation (Whipple triad), laboratory tests (hyperinsulinemia) and surgery. (3) The research either showed true positives (TR), false positives (FP), false negatives (FN), and true negatives (TN), or contained data from which these values could be calculated. The exclusion criteria were: (1) incomplete data, which could not be extracted for a contingency table; (2) non-original clinical studies; (3) duplicate studies; and (4) animal studies.

## Data extraction and quality evaluation

Two authors, Hao Wang and Ying Ba, independently extracted relevant information from the studies, including the authors' names, publication year, country, study design, gold standard, apparatus, methods, true and false positives and negatives. For disagreement settlement, if necessary, a third author (Qian Xing) joined the discussion to resolve the issue. There were some studies in which patients with islet hyperplasia also had a positive response to ASVS; if a case of islet hyperplasia was involved, the three authors of this study would discuss and resolve the problem of whether to include the study in the analysis. The Quality Assessment of Diagnostic Accuracy Studies (QUADAS-2) tool was used for quality evaluation of each included study [16, 17]. The tool is composed of two parts: the risk of bias (four domains: patient selection, index test, reference standard, and flow and timing) and concerns regarding applicability (three domains: patient selection, index test, and reference standard). If the answers to all signaling questions for a domain were "yes," the risk of bias was judged as "low." If any signaling question in a domain was "no," the risk of bias was judged as "high." The term "unclear bias" was used only if insufficient information was supplied.

## Statistical analysis

The included studies underwent quality evaluation using RevMan 5.3, followed by extracting the data from the included studies to calculate the true positive, false positive, false negative, and true negative values. Stata 14.0 software were used for statistical analysis. The bivariate effect model [18] was used to calculate the above data, followed by calculating positive likelihood ratio (PLR), negative likelihood ratio (NLR), diagnostic odds ratio (DOR), and summary receiver operating characteristic (SROC) [19]. The heterogeneity between the included studies was calculated using the Q test and $I^2$ test[20, 21]. For data with low heterogeneity ($I^2 < 50\%$), a fixed- effect model was used. A random-effect model was used for data with high heterogeneity ($I^2 \geq 50\%$). Meta-regression analysis of the DOR was used to find the sources of heterogeneity to analyze factors that affected heterogeneity, including experimental design, race, publication year, and patient type [22]. We assessed potential publication bias using the funnel plots of Deeks [23]. All P-values were two-sided, with $P < 0.05$ as the cut-off value for significance.

## Results

Fig 1 shows the results of the study selection process. The initial electronic search identified 405 articles, of which 162 were excluded due to duplication. A further 210 articles were excluded due to irrelevance (articles were removed after the titles and abstracts had been read). A total of 33 potentially eligible studies remained; after reviewing the full text of the articles and browsing the results, 23 articles were excluded because of incomplete data, lack of gold standards, or incomplete description of the trial (articles were removed after the full manuscript had been read).

Eventually, ten studies involving a total of 337 patients were included in the present systematic review and meta-analysis [24–33]. Table 1 lists the characteristics of the included studies.

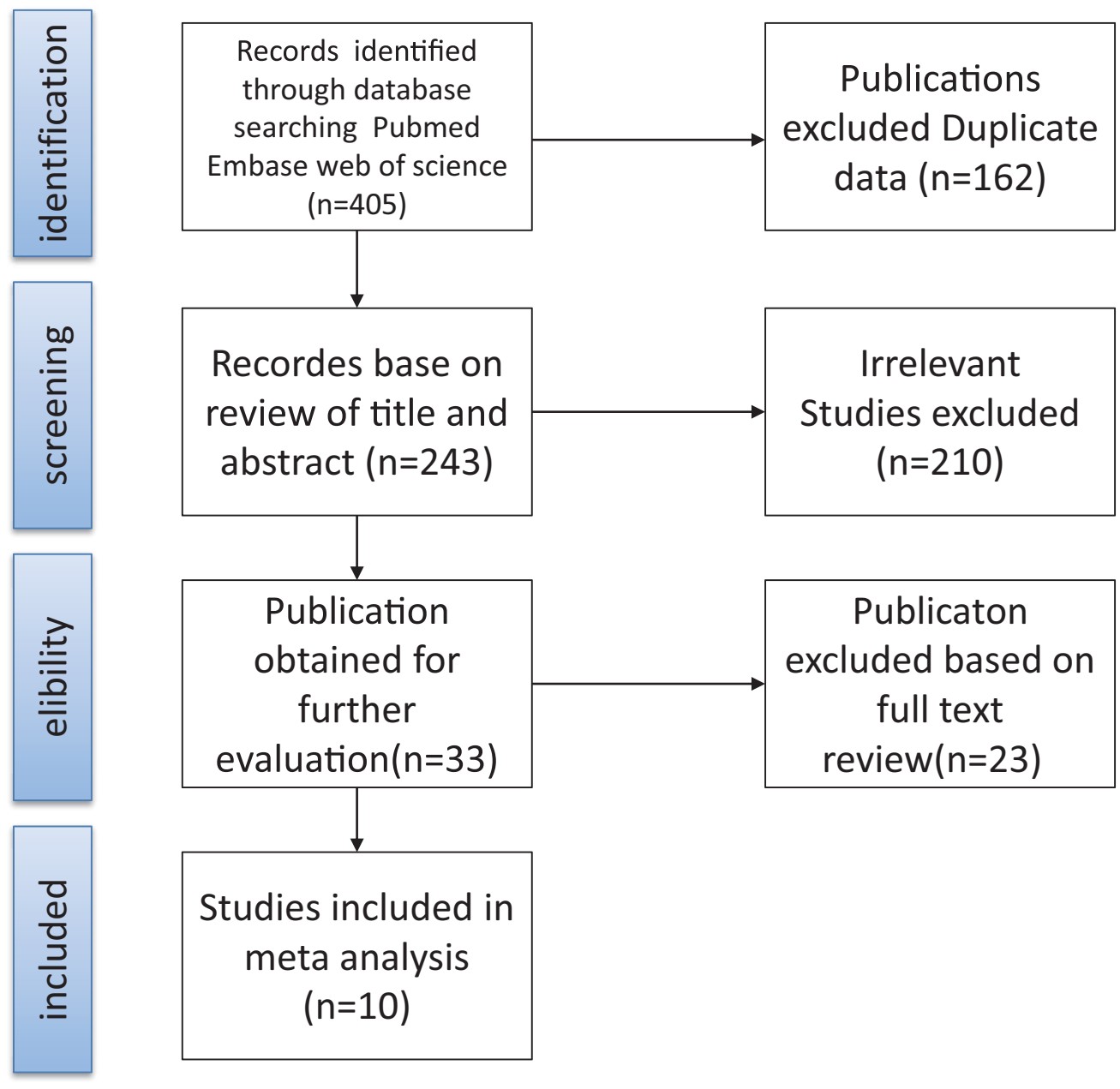

**Fig 1. Flowchart for obtaining data for the present meta-analysis.**

These studies had been published form 1992 through 2010. Six studies were conducted in Europe [25,26,28,30,32,33], two studies were conducted in the USA [24,29], and the remaining two studies conducted in China and New Zealand [27,31]. Five studies used a prospective design [24,25,26,28,31]; and five studies used a retrospective design [27,29,30,32,33]. Fig 2 shows the quality assessment results of the included studies.

Fig 3 presents the summarized results for sensitivity and specificity. The pooled sensitivity, specificity, PLR, and NLR were 0.93(95%confidence interval [CI]: 0.83–0.97), 0.86(95%CI: 0.75–0.93), 6.4(95%CI: 3.7–12.7), and 0.08(95%CI: 0.03–0.19), respectively. The pooled DOR

**Table 1. Characteristics of the studies included in the meta-analysis.**

| Author | Year | Country | Study design | Gold standard | Nesidioblastosis included | Calcium concentration(mEq/kg) | TP | FP | FN | TN |
|---|---|---|---|---|---|---|---|---|---|---|
| DoppmanJL[24] | 1995 | USA | Prospective | Clinic & surgery | No | 0.025 | 22 | 3 | 3 | 22 |
| OsheaD[25] | 1996 | UK | Prospective | Clinic & surgery | No | 0.025,0.00625 | 5 | 0 | 1 | 1 |
| DefreyneL[26] | 1998 | Belgium | Prospective | Clinic & surgery | No | 0.025 | 9 | 2 | 0 | 7 |
| JinZY[27] | 2002 | China | Retrospective | Pathology | Yes | 0.025 | 12 | 0 | 1 | 3 |
| WiesliP[28] | 2004 | Switzerland | Prospective | Pathology | No | 0.025 | 14 | 0 | 0 | 5 |
| GuettierJM[29] | 2009 | USA | Retrospective | Pathology | No | 0.025 | 38 | 2 | 5 | 43 |
| DruceMR[30] | 2010 | UK | Retrospective | Pathology | No | 0.025 | 17 | 4 | 6 | 23 |
| BraatvedtG[31] | 2014 | New Zealand | Prospective | Pathology | Yes | 0.0025,0.0065 | 16 | 3 | 0 | 18 |
| Morena J[32] | 2016 | France | Retrospective | Pathology | No | 0.025 | 10 | 1 | 1 | 10 |
| Moreno MP[33] | 2016 | Spain | Retrospective | Pathology | Yes | 0.025 | 18 | 7 | 0 | 5 |

of ASVS for insulinoma localization was 84(95%CI: 30–233, Fig 4). The SROC curve was 0.96 (95%CI: 0.94–0.97, Fig 5). The results of the heterogeneity studies (P = 0.00, $I^2$ = 80.17) were calculated using forests of the DOR, and threshold effects accounted for 59% of the sources of heterogeneity. Meta-regression analysis was conducted based on study design, publication year, enrolled population, ethnicity, calcium concentration, and patient type (i.e., with or without islet hyperplasia) (Fig 6). The results suggested that heterogeneity mainly derived from patient type. We determined the symmetry of the spots in the funnel plot, and Deeks' asymmetry test showed no evidence of publication bias (P = 0.56; Fig 7).

## Discussion

This article is the first meta-analysis to evaluate the diagnostic value of ASVS for insulinoma localization. Ten studies involving a total of 337 patients were included in the analysis. The sensitivity and specificity of ASVS for the localization of insulinoma were evaluated. The pooled sensitivity, specificity, and ROC were 0.93, 0.86, and 0.96, respectively, showing fairly good diagnostic efficiency. The high DOR suggested stronger discrimination ability for insulinoma localization. In addition, the area under the ROC curve was 0.96, indicating a high diagnostic accuracy rate. Therefore, ASVS is associated with high diagnostic value for preoperative localization of insulinomas, and ASVS can be combined with digital subtraction angiography and EUS to reduce the rate of misdiagnosis and improve the diagnostic value of preoperative localization of insulinoma.

Insulinoma is a functional tumor of the pancreas that can secrete insulin autonomously. ASVS does not locate insulinomas based on morphology, but does so functionally, which results in high sensitivity and specificity for the localization and diagnosis of the tumor. ASVS is highly suitable for the localization of such small but secretion-active tumors, and helps detect occult insulinomas [34]. For small and powerfully functional insulinomas, especially occult insulinomas, ASVS has higher accuracy for preoperative localization than other imaging examinations for preoperative localization. Therefore it is of great clinical value and makes it suitable as the gold standard for insulinoma localization. However, we distinguished insulinoma from islet cell hyperplasia in this article, which may have affected the specificity of our study. Insulinoma and islet cell hyperplasia differ in that neoplasms are absent from islet cell hyperplasia, but both secrete insulin, and surgery is indicated. Importantly, as we are using a meta-analysis/systematic review, we recognize that the results of the studies are highly dependent on the expertise and experience of the individual radiologists/institutions. Techniques

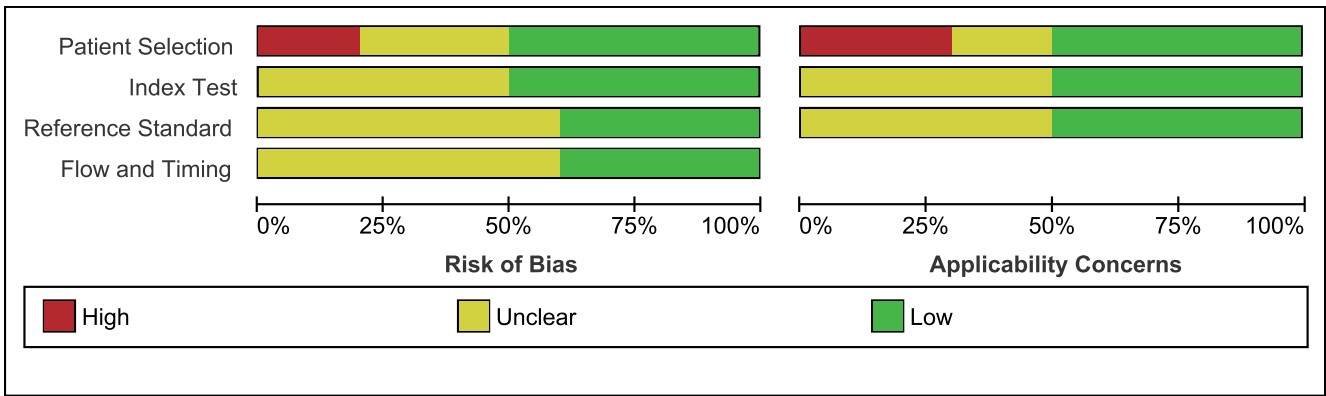

**Fig 2. Quality assessments of the included studies.**

may also have improved through the years, so not all the studies we included may be directly comparable.

ASVS may produce false positive results, The site suggested by ASVS may be inconsistent with the site confirmed by surgery, which may be attributed to vascular variation or mislocation resulting from persistent spasm after arterial intubation during surgery, leading to catheter withdrawal from the artery. If there are abnormal findings in routine examinations such as ultrasound, CT, and MRI, a corresponding arterial calcium stimulation in the suspected area can be performed first to reduce the possibilities of incorrect localization caused by calcium reflux due to arterial spasm. Different authors have used different calcium concentrations, and the effect of calcium concentration should also be further studied [35].

ASVS can locate an insulinoma regionally, but cannot locate its position precisely. Moreover, identifying a specific location is difficult if there are multiple insulinomas in the same region. If excessive insulin secretion is confirmed, but ASVS yields negative results in all areas of the pancreas, there may be an ectopic insulinoma.

Islet cell hyperplasia is usually difficult to detect on imaging examinations, while ASVS is very helpful for localization. The Mayo Clinic conducted a study to differentiate between insulinoma and islet hyperplasia by including 42 patients with insulinoma and 74 patients with islet hyperplasia and performing ASVS to observe insulin elevation after calcium gluconate. Of the 116 patients in that study, only one patient with islet hyperplasia had a non-significant increase in insulin. Insulin was elevated in the remaining 115 patients. In addition, the

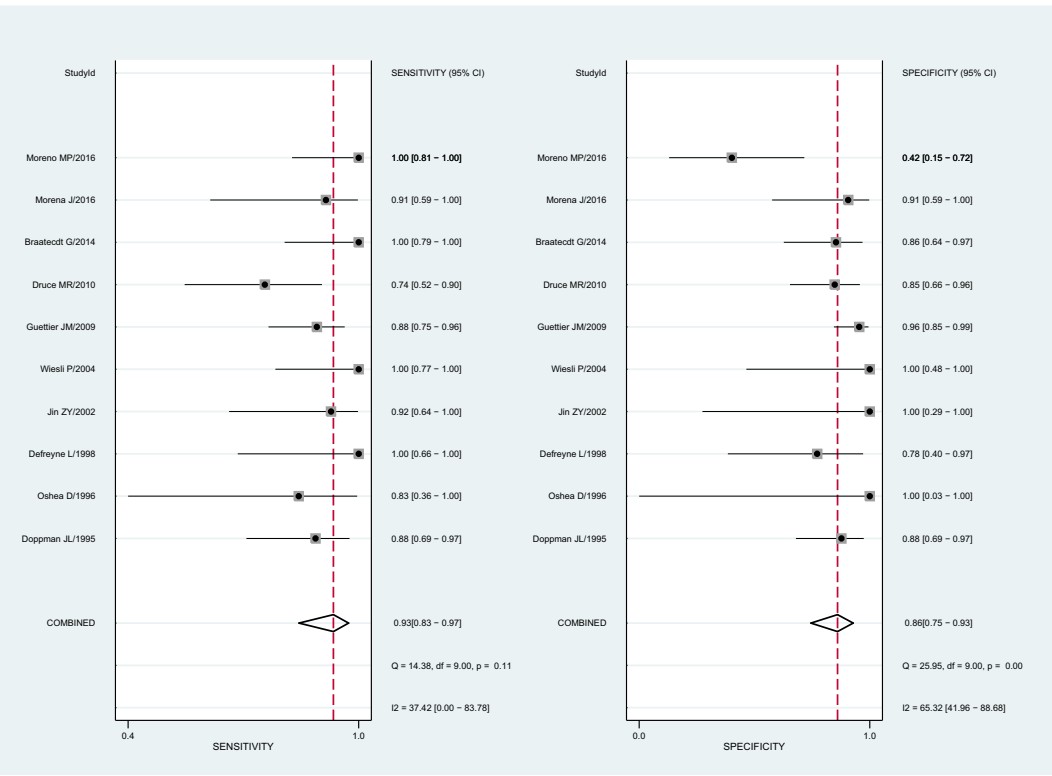

**Fig 3. Forest plots for sensitivity and specificity.**

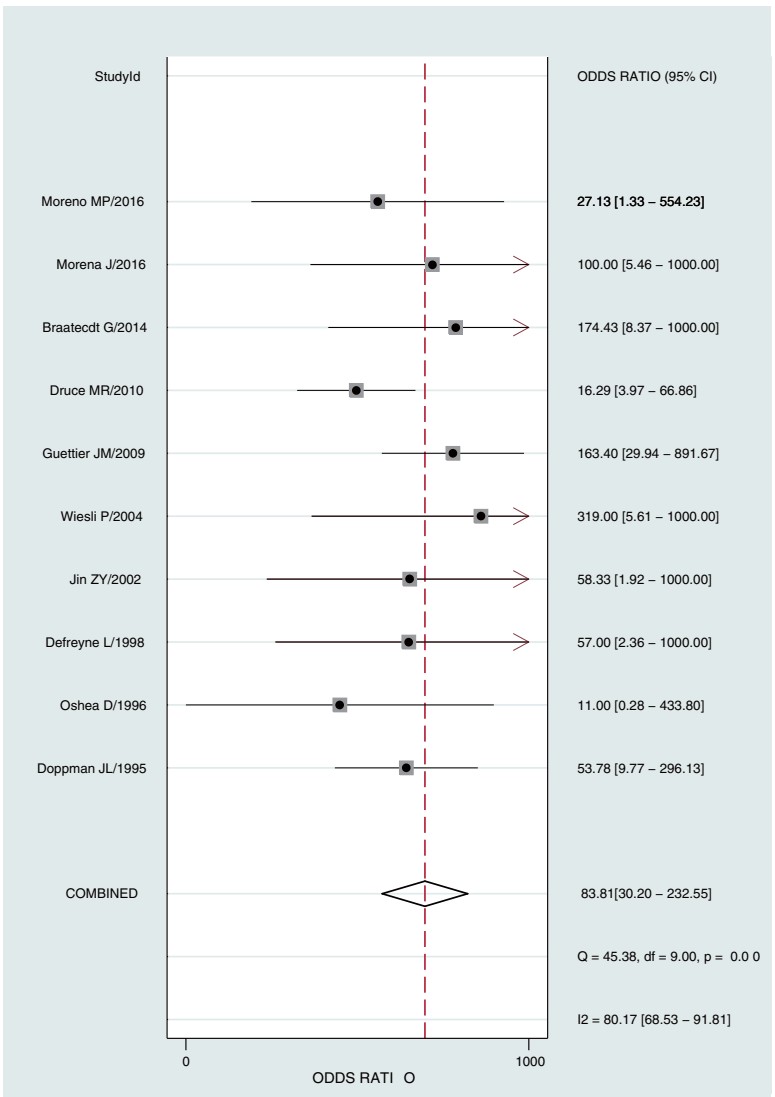

**Fig 4. Forest plot for DOR.**

elevation of insulin in the patients with insulinoma was significantly higher than in the patients with islet hyperplasia. In identifying the two, maximum hepatic venous insulin concentration (mHVI) cutoffs of > 91.5 and > 263.5 IU/mL were 95 and 100% specific for insulinoma, respectively. A 19-fold increase in hepatic venous insulin concentration (rHVI) over baseline was 99% specific for insulinoma [36]. The Mayo Clinic study suggested that ASVS was quite effective in identifying insulinoma and islet hyperplasia. Pereira *et al.* reported a case of nodular hyperplasia in the head of the pancreas. After calcium stimulation in the superior mesenteric artery and gastroduodenal artery, the insulin levels in the blood increased by two- and seven-fold, respectively. These levels increased continuously, showing a response curve different from that of typical insulinoma [37]. In a case of mild islet cell hyperplasia in the tail and body of the pancreas, insulin secretion peaks appeared after calcium stimulation in the proximal and distal splenic artery, but they were less than twice the baseline values, The height of such peaks is considered related to the degree of hyperplasia [27]. Due to the limited number

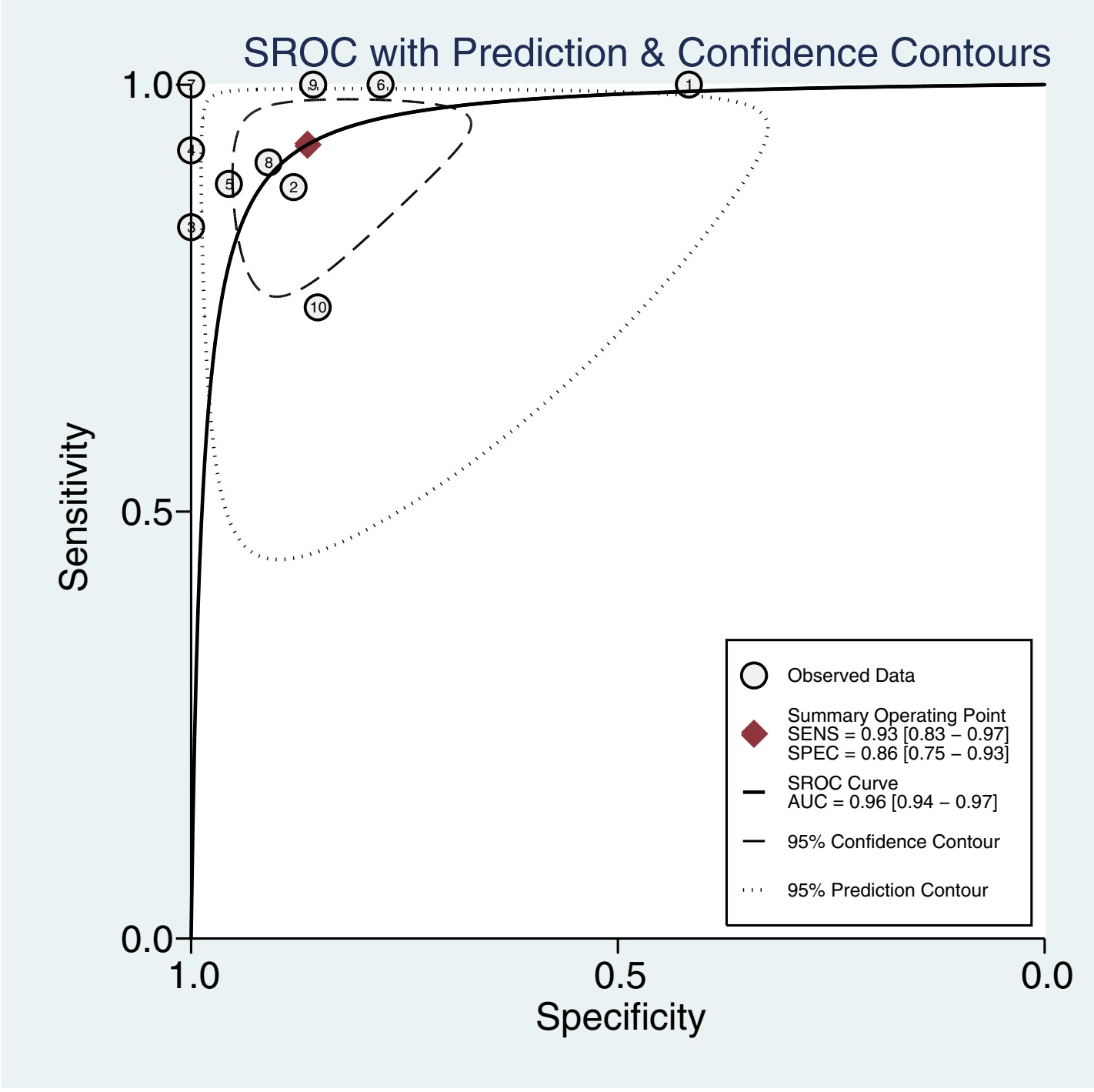

**Fig 5. Area under the ROC curve.**

of cases, the effect of ASVS on the insulin secretion curve in islet cell hyperplasia requires further study.

While ASVS has its advantages in insulinoma localization, other methods for the localization of insulinoma also have advantages. SPECT/CT imaging is a very promising technique for

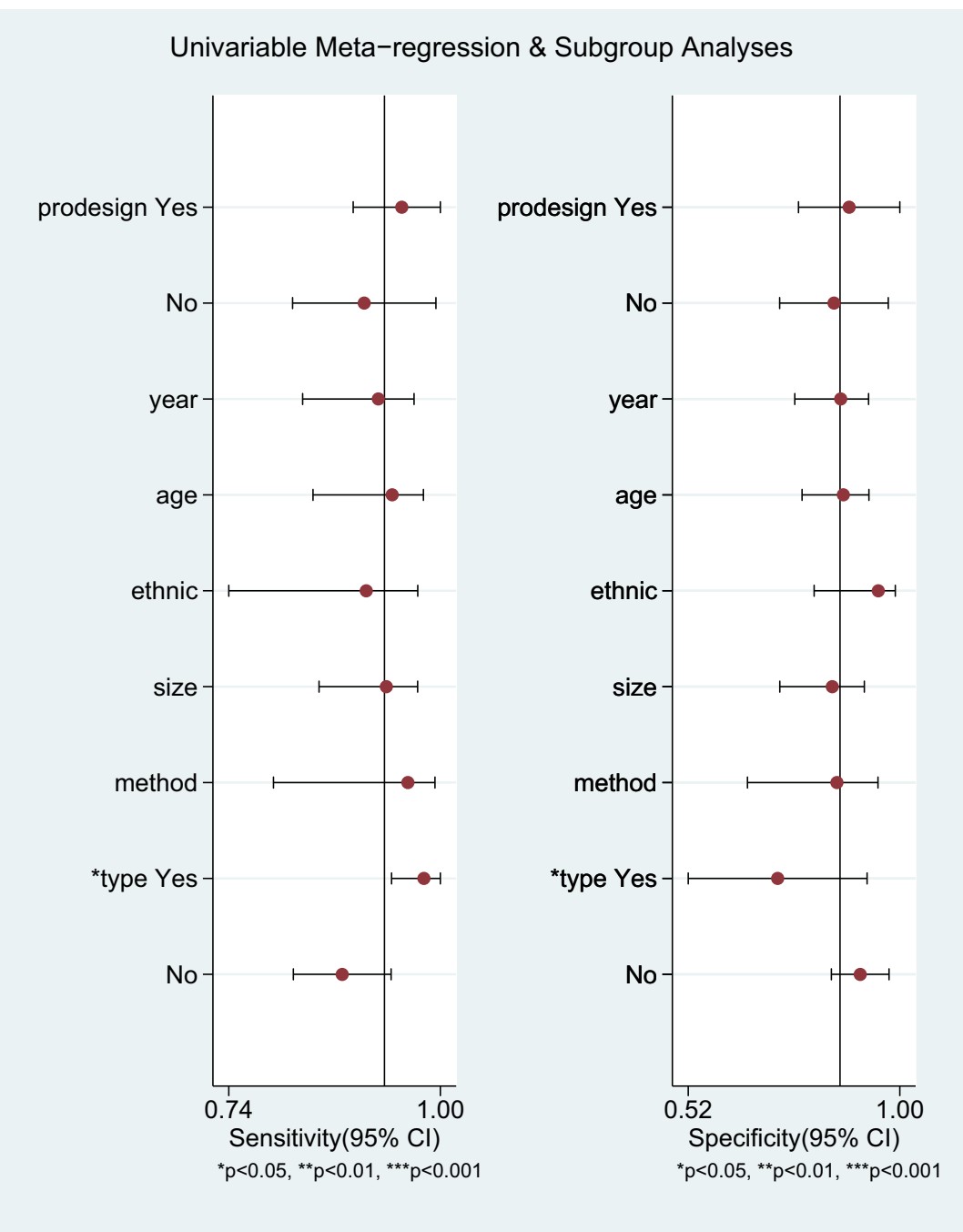

**Fig 6. Meta-regression analysis for DOR.**

insulinoma localization, and uses radionuclide-labeled glucagon-like peptide-1 receptor ago-nist (GLP-1RA) to bind to glucagon-like peptide-1 receptor (GLP-1R) for radionuclide imag-ing. Because there is high expression of GLP-1R in insulinomas, its binding to the GLP-1RA probe labeled by different radionuclides can identify the tumor location. Generally, analogues of Exendin-4 are used as the GLP-1RA probes, labeled by radionuclides such as [111]In, [99]Tcm, and [68]Ga. Christ *et al.* performed preoperative SPECT/CT imaging with [111]In-DOTA-Lys[40]-

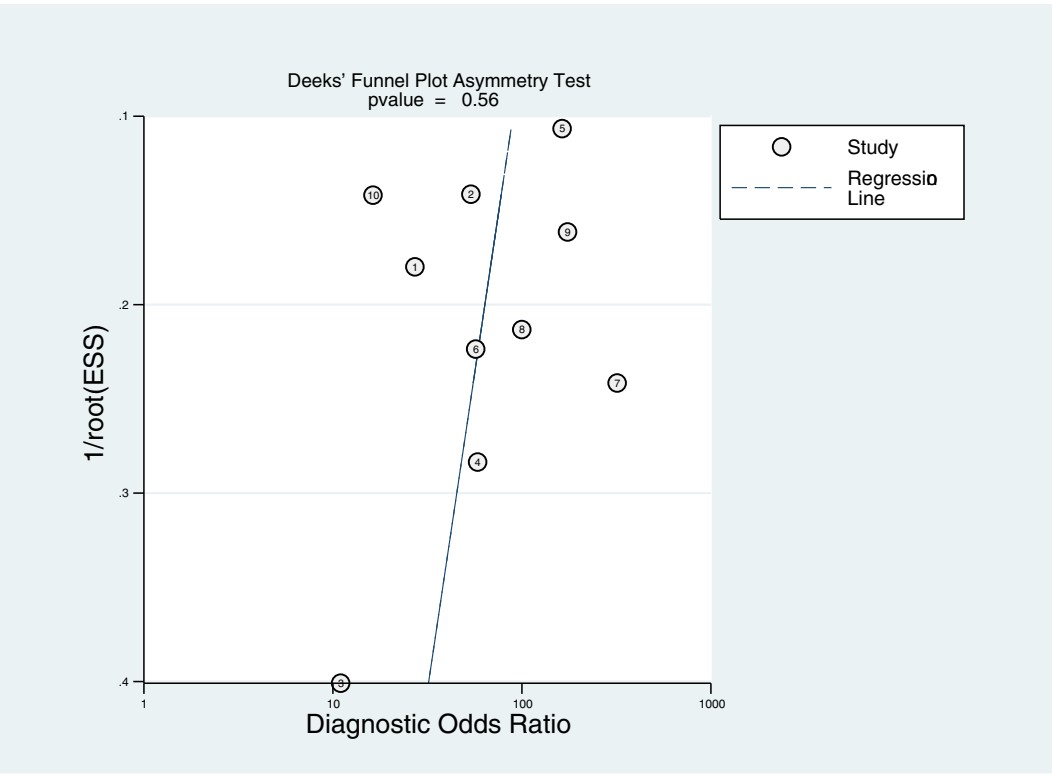

**Fig 7. Deeks' plot for ASVS for localization of insulinoma.**

Exendin-4 in six patients suspected of insulinoma, and observed definite tumor lesions in all the cases, with lesion diameters of 9 to 18 mm. Five of the six cases were diagnosed with pancreatic insulinomas, and the remaining one with ectopic insulinoma, all of which were confirmed as benign by postoperative pathology [38]. Sowa-Staszczak *et al.* performed radionuclide imaging with $^{99}$Tcm-HYNIC-Lys$^{40}$-Exendin-4 in 11 patients with persistent hyperinsulinemia and hypoglycemia (eight with benign insulinoma undetected by CT scans, two with malignant insulinoma, and one with islet cell hyperplasia). The radionuclide imaging successfully detected the lesions in all the cases with benign insulinoma, but failed in one case with malignant insulinoma [39]. A study showed that only 36% of malignant insulinomas express GLP-1R, resulting in a low detection rate of malignant insulinoma by GLP-1R imaging [40]. Luo *et al.* conducted a prospective study in which $^{68}$Ga-NOTA-Exendin-4 imaging detected 42 positive cases from 43 patients with insulinoma, with a sensitivity of 97.7% [41]. Therefore, the imaging of insulinoma using radionuclide-labeled GLP-1RA has high sensitivity for insulinoma diagnosis, and the specific binding of GLP-1RA to GLP-1R leads to a significant advantage in insulinoma localization, showing promise for use in preoperative and intraoperative localization of insulinoma, as well as its postoperative follow-up.

According to our meta-analysis, after a definite qualitative diagnosis of insulinoma, ASVS should be performed for localization. If an imaging examination yields negative results, ASVS may still be useful for insulinoma localization and for detecting occult insulinoma. This suggests that more patients with islet cell hyperplasia and non-insulinoma should be included in studies on the value of ASVS in insulinoma localization and diagnosis. Standards should be established for the dosage of calcium used for stimulation, and more studies should be conducted on the different manifestations of insulinoma and islet cell hyperplasia.

The strengths of this study are that we followed a standard protocol and used a comprehensive search strategy. We also calculated the bivariate random effects model and performed hierarchical SROC analyses. In addition, the pooled sample size was large so the present findings are more robust than any individual study. Lastly, heterogeneity was explored using meta-regression analysis. This indicated that the heterogeneity observed was due to different patient types.

The limitations of this study are as follows: First, data on the patients' characteristics were not available, which might have affected the diagnostic value of ASVS. Second, we used summarized data, which restricted detailed analysis. Finally, the present study is based on published studies, and publication bias is an inevitable problem.

The present study is the first meta-analysis to determine the diagnostic value of ASVS for localization of insulinomas, and suggests that ASVS is associated with high diagnostic value for insulinoma localization. These findings require validation in further, large-scale prospective studies to evaluate the diagnostic value of ASVS in patients with specific characteristics.

## Supporting information

**S1 File. PRISMA checklist.**
(DOC)

**S2 File. Included studies.**
(ZIP)

## Author Contributions

**Conceptualization:** Hao Wang, Ying Ba.

**Software:** Hao Wang.

**Writing – original draft:** Hao Wang, Ying Ba, Qian Xing, Run-Ce Cai.

**Writing – review & editing:** Hao Wang, Ying Ba.

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
