## [Decision Letter · Decision Letter 0]

28 Jun 2019

PONE-D-19-14887

Diagnostic value of ASVS for insulinoma localization: A systematic review and meta-analysis

PLOS ONE

Dear Dr. Hao Wang,

Thank you for submitting your manuscript to PLOS ONE. After careful consideration, we feel that it has merit but does not fully meet PLOS ONE’s publication criteria as it currently stands. Therefore, we invite you to submit a revised version of the manuscript that addresses the points raised during the review process.

ACADEMIC EDITOR: Although it is of interest, the reviewers have raised a number of points which we believe major modifications are necessary to improve the manuscript, taking into account the reviewers' remarks.  

We would appreciate receiving your revised manuscript by Aug 12 2019 11:59PM. To enhance the reproducibility of your results, we recommend that if applicable you deposit your laboratory protocols in protocols.io, where a protocol can be assigned its own identifier (DOI) such that it can be cited independently in the future. For instructions see: http://journals.plos.org/plosone/s/submission-guidelines#loc-laboratory-protocols

We look forward to receiving your revised manuscript.

Kind regards,

Wisit Cheungpasitporn, MD, FACP

University of Mississippi Medical Center

Twitter: @wisit661 Email: wcheungpasitporn@gmail.com 

Academic Editor

PLOS ONE

Journal Requirements:

2. We noticed you have some minor occurrence of overlapping text with the following previous publication, which needs to be addressed: Wang, Hao, et al. "Diagnostic value of endoscopic ultrasound for insulinoma localization: A systematic review and meta-analysis." PloS one 13.10 (2018): e0206099. In your revision ensure you cite all your sources (including your own works), and quote or rephrase any duplicated text outside the methods section. Further consideration is dependent on these concerns being addressed.

3. Please provide an institutional email address for each author.

4. Thank you for stating the following financial disclosure: NO

Please provide an amended Funding Statement that declares *all* the funding or sources of support received during this specific study (whether external or internal to your organization) as detailed online in our guide for authors at http://journals.plos.org/plosone/s/submit-now.  

Please state what role the funders took in the study.  If any authors received a salary from any of your funders, please state which authors and which funder. If the funders had no role, please state: "The funders had no role in study design, data collection and analysis, decision to publish, or preparation of the manuscript."

Reviewers' comments:

Reviewer's Responses to Questions

**Comments to the Author**

1. Is the manuscript technically sound, and do the data support the conclusions?

Reviewer #1: Partly

Reviewer #2: Partly

Reviewer #3: Partly

2. Has the statistical analysis been performed appropriately and rigorously? 

Reviewer #1: Yes

Reviewer #2: Yes

Reviewer #3: I Don't Know

3. Have the authors made all data underlying the findings in their manuscript fully available?

Reviewer #1: No

Reviewer #2: No

Reviewer #3: No

4. Is the manuscript presented in an intelligible fashion and written in standard English?

Reviewer #1: No

Reviewer #2: No

Reviewer #3: Yes

5. Review Comments to the Author

Reviewer #1: 1. Pathology diagnosis prior to year of 2000 likely are not accurate. Please perform sensitivity analysis excluding those 1990s studies.

2. Need to provide supplementary data on pathology in each study for the diagnosis of insulinoma.

3. Manuscript is very poor written in English. For example, "Chian" in table is not correct country. "Zealand" is not correct. There are also many errors throughout manuscripts.

Reviewer #2: This meta-analysis has not been registered online. Please add this point in the limitation.

Who are two independent investigators?

It will be better to show kappa for the selection and data extraction. Please show the data of kappa of agreement during the systematic searches. How disagreements were solved during the systematic search among two independent reviewers?

Search terms in PubMed and Embase are different. Please attach syntax used in each database as supplementary.

Please make the data for this review publicly available, possibly through the Open Science Framework (osf.io). Items to include: list of excluded studies, commands for statistical analysis, spreadsheets or data used for the meta-analyses, etc. Making data publicly available will promote the reproducibility of the review and is best practices for systematic reviews and meta-analyses.

Authors should discuss the reason of heterogeneity.

Please include I2 in the abstract.

It is “fixed-effects model and random-effects model”, not “fixed-effects model and random effect model”.

Revision of the English writing is strongly recommended. There are many grammar errors and misspells.

Reviewer #3: The review is concise and relatively easy to read. However, there are several important points which should be addressed.

I was unable to determine if all the relevant articles had been included for review. Two studies which could be included also would be

1. Placzkowski, Vella et al: JCEM 2009;94:1069 which describes trends in the use of success of the technique from the Mayo Clinic.

2.Thompson, Vella et al:JCEM 2015;100:4189

Also, the use of meta-analysis/ systematic review should also recognise that the results of the studies would also be highly dependent on the expertise and experience of the individual radiologist/ institution. Techniques presumably also have improved through the years- which should also be taken into account.

Mention should also be made of the potential role of GLP-RA labelled nuclear scans which in some hands appear to provide greater sensitivity and specificity than the more invasive tests.

I presume that the techniques used are sound otherwise.

6. PLOS authors have the option to publish the peer review history of their article (what does this mean?). If published, this will include your full peer review and any attached files.

Reviewer #1: No

Reviewer #2: No

Reviewer #3: No

---

## [Author Response · Author response to Decision Letter 0]

13 Aug 2019

1 We have read your model carefully and revised the article according to the model.

2 For the overlapping section, we have modified it.

3 Sorry, we have not institutional email address.

4 we have not any funding or sources of support.

1 We write this meta-analysis in strict accordance with the standard format of meta-analysis, search the relevant literature correctly, extract data, and draw data and conclusions through correct statistical analysis.

2 Our statistics are carried out in accordance with the normal diagnostic meta-analysis, strictly according to the input and output criteria into the relevant research. After extracting the data, according to TP FP FN TN, using STATA Software 14.0 for meta-analysis, using random effect model analysis.

3 We provide the full text of the study and present the data in tabular form. We have no restrictions on the sharing of documentary data.

4 English is not our mother tongue, but after we finish writing the article, we invite English experts to translate and polish it.

5 Reviewer #1: 1. As you said, pathology diagnosis prior to year of 2000 likely are not accurate. But we carefully read the three studies published 2000 years ago. These three studies were published in rigorous and regular clinical journals. The diagnosis of insulinoma is clear through clinical manifestations(Whipple's triad syndrome), biochemical tests(Hyperinsulinemia), surgery and so on. We believe that the diagnosis is reliable, so the data of these three studies are still retained.

 2. We provided the pathological results of each study.

 3. Thank you ,teacher. We know the manuscript is very poor written in English.But we did our best to improve it, and invite native language editors to translate and polish it in English.

 Reviewer #2: Two independent investigators refer to Hao Wang and Ying Ba, Later, the article was also marked.

 When we collect data, we have strict requirements for the operation process and set the Kappa standard.

 Search syntax: PubMed indexed the collected data according to the medical thesaurus. Thesaurus retrieval can improve the efficiency of literature retrieval. MeSH network version is integrated in PubMed database. The application of MeSH Database can determine the standard search terms, view the annotations of word meanings, tree structure tables, view the sub-headings and extensions. PubMed search supports Boolean logic operations, AND, OR, NOT. It can be retrieved by AND, OR and NOT combined subject words and free words. Choosing appropriate subject words and free words retrieval can be more comprehensive and accurate. So we use the search strategy in PubMed: “arterial calcium stimulation” AND “insulinoma” [MeSH]

Embase provides an Independent Thesaurus Retrieval System (EMTREE). EMTREE thesaurus is one of the most powerful retrieval tools of Embase. EMTREE contains more than 54,000 preferred terms, more than 2.1 million synonyms and multi-level tree structure. The use of EMTREE keyword retrieval can improve the comprehensiveness and accuracy of literature retrieval.So we use the search strateg in Embase: Emtree term–exploded = (insulinoma AND arterial calcium stimulation)

 During the systematic search when we meet disagreement, we negotiate. For example, Ying Ba want include a Chinese article about ASVS, but I don't think it's right because the article published in an informal magazine. I can't confirm its scientific nature. Later Ying Ba agreed with me.

 We already discuss the reason of heterogeneity.

 We already include I2 in the abstract.

 English is not our first language, but after we finish writing the article, we invite English experts to translate and polish it.

 Reviewer #3:Thank you for your comments. We actually notice the two article before.

 We have read these two articles carefully and thought that these two articles are very helpful to study the diagnostic value of ASVS in localizing insulinoma, however, in "Placzkowski, Vella et al." 's article, ASVS has only mention positive diagnostic rate. There is no specific data, and there is no data of non-insulinoma control group, so we have not included in this study.

 For Thompson, Vella et al, they studied the differential diagnosis of insulinoma and islet hyperplasia by ASVS. We also thought it's a very good study.However,we believe that there is a lack of data for the control group of non-insulinoma and non-islet hyperplasia. Because in the study of ASVS, besides the positive rate of ASVS in insulinoma and islet hyperplasia, the negative rate of non-insulinoma and islet hyperplasia is also important, In theory, both insulinoma and islet proliferation should have insulin secretion after calcium stimulation. So we have not included this study. However, in the discussion, we quoted Thompson, Vella 's article. Because this article is indeed a good study of ASVS in the differential diagnosis of insulinoma and islet hyperplasia.

 As you suggest, We already mention the potential role of GLP-RA labelled nuclear scans which in some hands appear to provide greater sensitivity and specificity than the more invasive tests.

6 We agree to our identity to be public for this peer review

---

## [Decision Letter · Decision Letter 1]

2 Sep 2019

PONE-D-19-14887R1

Diagnostic value of ASVS for insulinoma localization: A systematic review and meta-analysis

PLOS ONE

Dear Hao Wang,

Thank you for submitting your manuscript to PLOS ONE. After careful consideration, we feel that it has merit but does not fully meet PLOS ONE’s publication criteria as it currently stands. Therefore, we invite you to submit a revised version of the manuscript that addresses the points raised during the review process.

ACADEMIC EDITOR: Our expert reviewers still have raised a number of points which we believe major modifications are necessary to improve the manuscript, taking into account the reviewers' remarks below.

We would appreciate receiving your revised manuscript by Oct 17 2019 11:59PM. To enhance the reproducibility of your results, we recommend that if applicable you deposit your laboratory protocols in protocols.io, where a protocol can be assigned its own identifier (DOI) such that it can be cited independently in the future. For instructions see: http://journals.plos.org/plosone/s/submission-guidelines#loc-laboratory-protocols

We look forward to receiving your revised manuscript.

Kind regards,

Wisit Cheungpasitporn, MD, FACP

Academic Editor

PLOS ONE

Reviewers' comments:

Reviewer's Responses to Questions

**Comments to the Author**

1. If the authors have adequately addressed your comments raised in a previous round of review and you feel that this manuscript is now acceptable for publication, you may indicate that here to bypass the “Comments to the Author” section, enter your conflict of interest statement in the “Confidential to Editor” section, and submit your "Accept" recommendation.

Reviewer #1: All comments have been addressed

Reviewer #2: All comments have been addressed

Reviewer #3: (No Response)

2. Is the manuscript technically sound, and do the data support the conclusions?

Reviewer #1: Yes

Reviewer #2: Yes

Reviewer #3: Partly

3. Has the statistical analysis been performed appropriately and rigorously? 

Reviewer #1: Yes

Reviewer #2: Yes

Reviewer #3: I Don't Know

4. Have the authors made all data underlying the findings in their manuscript fully available?

Reviewer #1: Yes

Reviewer #2: Yes

Reviewer #3: Yes

5. Is the manuscript presented in an intelligible fashion and written in standard English?

Reviewer #1: Yes

Reviewer #2: Yes

Reviewer #3: Yes

6. Review Comments to the Author

Reviewer #1: It appears that all comments have been appropriately responded to. I have no further comments and recommend publication.

Reviewer #2: Much improved manuscript from prior version. The paper, in its revised form, is suitable for publication

Reviewer #3: The authors have improved the manuscript with corrections to grammatical mistakes seen in the first submission. There is an attempt to discuss the findings in more detail and to answer the queries of the reviewers. However, there still needs to be acknowledgement of the wide range of expertise in the use of this investigation which would have significant implications on the results. GLP 1 receptor imaging needs to be included in the discussion in some detail as a noninvasive technique to locate the insulinoma. There is also a relative paucity of published studies overall with less positive experiences likely not submitted for publication.

7. PLOS authors have the option to publish the peer review history of their article (what does this mean?). If published, this will include your full peer review and any attached files.

Reviewer #1: No

Reviewer #2: No

Reviewer #3: No

---

## [Author Response · Author response to Decision Letter 1]

21 Sep 2019

responds to academic editor

1. We have deposited my laboratory protocols in protocols.io. We get the DOI: dx.doi.org/10.17504/ protocols.io.7imhkc6.

2. We have no change to my financial disclosure.

3. We have upload the revision of my manuscript according to your direction.

Responds to the reviewers:

1. Thank you. We have addressed reviewers’ comments raised in a previous round of review.

2. The manuscript was written in accordance with normal criteria of standard diagnostic meta analysis. So We think it is technically sound, and the data do support the conclusions.

3. The manuscript comply with the Standard Writing of Meta-analysis. And We think statistical analysis been performed appropriately and rigorously.

4. All data underlying the findings described in the manuscript fully available without restriction. We already deposited the meta analysis and the relevant data and step to a public repository. We got the DOI: dx.doi.org/10.17504/ protocols.io.7imhkc6.

5. After we have finished the paper, we invited several mother tongue editor to polish the language.

6. Dear teacher,Thank you very much for your instruction. We also think that the wide range of expertise in the use of this investigation which would have significant implications on the results. For the role of GLP 1 receptor imaging in the localization of insulinoma, we also searched a lot of literature, and made a more detailed discussion. 

7. Thanks for the peer review. We learnt a lot.

---

## [Decision Letter · Decision Letter 2]

25 Oct 2019

Diagnostic value of ASVS for insulinoma localization: A systematic review and meta-analysis

PONE-D-19-14887R2

Dear Dr. Hao Wang,

We are pleased to inform you that your manuscript has been judged scientifically suitable for publication and will be formally accepted for publication once it complies with all outstanding technical requirements.

With kind regards,

Wisit Cheungpasitporn, MD, FACP

Academic Editor

PLOS ONE

Additional Editor Comments:

I want to commend the authors on their superb efforts to revise the manuscript according to all reviewers’ suggestions. The quality of the manuscript has improved substantially.

Reviewers' comments:

Reviewer's Responses to Questions

**Comments to the Author**

1. If the authors have adequately addressed your comments raised in a previous round of review and you feel that this manuscript is now acceptable for publication, you may indicate that here to bypass the “Comments to the Author” section, enter your conflict of interest statement in the “Confidential to Editor” section, and submit your "Accept" recommendation.

Reviewer #1: All comments have been addressed

Reviewer #2: All comments have been addressed

2. Is the manuscript technically sound, and do the data support the conclusions?

Reviewer #1: Yes

Reviewer #2: Yes

3. Has the statistical analysis been performed appropriately and rigorously? 

Reviewer #1: Yes

Reviewer #2: Yes

4. Have the authors made all data underlying the findings in their manuscript fully available?

Reviewer #1: Yes

Reviewer #2: Yes

5. Is the manuscript presented in an intelligible fashion and written in standard English?

Reviewer #1: Yes

Reviewer #2: Yes

6. Review Comments to the Author

Reviewer #1: It appears that all comments have been appropriately responded to. I have no further comments and recommend publication.

Reviewer #2: Very important research, much needed. The investigators have addressed all comments and concerns appropriately. I have no additional concerns.

7. PLOS authors have the option to publish the peer review history of their article (what does this mean?). If published, this will include your full peer review and any attached files.

Reviewer #1: No

Reviewer #2: No

---

## [Editor Report · Acceptance letter]

29 Oct 2019

PONE-D-19-14887R2 

Diagnostic value of ASVS for insulinoma localization: A systematic review and meta-analysis 

Dear Dr. Wang:

I am pleased to inform you that your manuscript has been deemed suitable for publication in PLOS ONE. Congratulations! Your manuscript is now with our production department. 

With kind regards,

on behalf of

Dr. Wisit Cheungpasitporn 

Academic Editor

PLOS ONE